# Consumer Information Search in Live-Streaming: Product Involvement and the Moderating Role of Scarcity Promotion and Impulsiveness

**Yuanyuan Guo [1], Xiaoting Chen [1] and Chaoyou Wang [2],***

[1]   School of Maritime Economics and Management, Dalian Maritime University, Dalian 116026, China; guoyuanyuan@dlmu.edu.cn (Y.G.); cxt_stu@163.com (X.C.)
[2]   School of Management Science and Engineering, Dongbei University of Finance and Economics, Dalian 116025, China
*   Correspondence: wangchaoyou@dufe.edu.cn

**Abstract:** While the literature acknowledges the impact of product involvement on consumer information search, little work discerns the boundary conditions of product involvement on information searching. Product involvement reflects an individual's interest in an object because of its inherent values, necessity, and interest. This study investigates the combined moderating role of limited-quantity scarcity and personal impulsiveness in the relationship between product involvement and information search behavior. A survey experiment with 402 participants was conducted to test this hypotheses. The experiment in this study used a 2 (cognitive involvement: high vs. low) $\times$ 2 (affective involvement: high vs. low) $\times$ 2 (limited-quantity scarcity: high vs. low) between-subjects design. The results provided strong evidence that (1) cognitive involvement is positively associated with online information search, whereas affective involvement is not associated with online information search; (2) limited-quantity scarcity significantly weakens the impact of cognitive involvement on online information search, but it does not have an interaction effect with affective involvement on online information search; and (3) the three-way interaction among product involvement (i.e., cognitive involvement and affective involvement), limited-quantity scarcity, and impulsiveness on consumer information search is significant. This study extends the current information searching studies by uncovering personal impulsiveness and limited-quantity scarcity as boundary conditions that influence the effects of cognitive involvement and affective involvement on consumer information search. The findings can help stakeholders promote the sustainability of e-commerce live-streaming in practice.

**Keywords:** online information search; limited-quantity scarcity; impulsiveness; cognitive involvement; affective involvement

## 1. Introduction

E-commerce live-streaming is a business model in which retailers, influencers, or celebrities sell a variety of products and services via online video streaming where the presenter demonstrates and discusses the offering and answers audience questions in real-time [1]. Combined with e-commerce and live-streaming, e-commerce live-streaming is a form of social commerce and has created a new shopping method. For example, Li Jiaqi sold USD 19 billion worth of goods on Single's Day in 2021, attracting almost 250 million viewers during the event [2]. E-commerce live-streaming fosters authenticity, visualization, interactivity, and entertainment, which could trigger different consumers' behavioral patterns during the decision-making process, such as information search, purchase behavior, consumer engagement, and even impulse purchasing [3–6].

While the focus of e-commerce live-streaming research so far has been on purchase behavior and impulse purchasing [3,7,8], research on pre-purchase information searching is scant. Information searching is a consequential early stage in the consumer decision-making

process, and consumers are active search agents that commonly depend on information to make decisions. Consumers' information searching in e-commerce live-streaming might differ from the conditions in traditional online shopping. In traditional online shopping, consumers have enough time to search the information concerning a particular product because they have a purchasing plan. However, in a live-streaming context, consumers might have not planned to buy anything, but they suddenly want to buy a product in the process of watching live-streaming. In such a situation, instantaneous information searching is needed, and consumers' perception of time pressure is more intense, especially under the conditions of streamers' scarcity promotion strategy. Therefore, it is worth discussing consumers' information search patterns and the affecting factors in e-commerce live-streaming.

Product involvement refers to an individual's interest in an object because of its inherent values, necessity, and interest [9]. It can be categorized into two types, namely cognitive involvement and affective involvement [10]. Cognitive involvement emphasizes reasoning and factual information, which are induced by utilitarian or cognitive motives. Affective involvement focuses on emotion and mood, which derive from value-expressive or affective motives. The consumer behavior literature has often emphasized the link between product involvement and information search and found that cognitive involvement is positively related to information searching, while affective involvement is not associated with information searching [11]. However, these studies did not uncover the reasons why affective involvement is not related to information searching. Schaefer et al. [11] and Hansen [12] call for more research to explore other variables that may moderate the relationship between involvement and information searching. This study answers this call by focusing on the boundary conditions of cognitive and affective involvement on information searching.

Although some factors, such as product characteristics, environmental factors, and individual traits work together to enable consumer information searching, they are often examined separately in the literature [13–15]. It is necessary to take account of the simultaneous interaction effects among these factors. Specifically, because consumers' attitudes and psychological features of decision making vary with level of involvement, their perceptions of scarcity promotion strategy will also differ, indicating that product involvement may interact with scarcity promotion strategy in e-commerce live-steaming. Therefore, this study analyzes the moderating effect of limited-quantity scarcity on the relationship between product involvement and information searching. Meantime, as an environmental cue, the benefits of scarcity promotion strategy may depend on consumers' impulsiveness, which influences consumers' perceptions to the scarcity promotion. Although past works have shown the significance of personal impulsiveness and scarcity promotion strategy in impulse buying [16–18], they have neglected the possible effect of any interaction between scarcity promotion and impulsiveness, especially their interaction effects on the link between product involvement and information searching. Thus, to better reflect the reality of business, this study investigates changes in the effect of limited-quantity scarcity with consumers' high and low impulsiveness.

This paper makes several contributions to the consumer behavior literature in e-commerce live-streaming. First, rather than repeat investigations into impulse purchasing and purchase intentions, this study extends the consumer behavior literature by investigating the pre-purchase information search behavior and its determinants. Second, this study contributes to the information search literature [11,13] by discerning boundary conditions for the effects of product involvement on information searching. This is important because, until recently, it was unclear why affective involvement was not associated with the level of information searching. Third, although product involvement, limited-quantity scarcity, and personal impulsiveness have each previously shown a relationship with consumer decision making in isolation [13,14], the combination of them is a valuable extension of prior research to help explain how they operate in unison to affect consumer information search. In terms of practical contributions, first, the suggestions for retailers and live-streamers can help them understand the factors which influence consumers' information searching

in live-streaming, thereby delivering relevant and targeted information to consumers for decision making and promoting the sustainable and healthy development of live-streaming. Second, the findings of this study can help consumers reasonably examine their information searching behavior and establish the concept of sustainable consumption.

## 2. Theoretical Background

### 2.1. Consumer Behaviors in E-Commerce Live-Streaming

E-commerce live-streaming research mainly focuses on consumers' purchase intention and impulse purchases. Based on IT affordance theory, Sun et al. [3] demonstrate that visibility affordance, metavoicing affordance, and guidance shopping affordance are associated with consumers' purchasing intentions. Lu and Chen [19] argue that streamers' physical characteristics can help decrease product certainty and gain consumers' trust, which subsequently influences consumers' purchase intention. Similarly, Xu et al. [20] suggest that streamers' attractiveness, information quality, and para-social imagination in the live-streaming context is significantly associated with emotional energy, which subsequently has a significant impact on consumer impulse purchases.

Additionally, a few live-streaming studies have focused on audiences' watching intentions or streamers' continuous broadcasting intentions. Based on the social identify theory, Hu et al. [21] indicate that audience's identification with streamers and audience groups positively influence their watching intentions. Chen and Lin [22] suggest that the four factors affecting audiences' watching live-streaming intentions are entertainment, flow, social interaction, and endorsement. Drawing upon the self-determination theory, Zhao et al. [23] argue that streamers' intrinsic and extrinsic motivation significantly influences their performance expectations, which then influence their continuance broadcasting intentions.

When making purchase decisions, consumers' decision-making process can be divided into five stages, namely the need for problem recognition, information search, evaluation of alternatives, purchase decision, and post-purchase evaluation [24]. Although previous studies have greatly improved our understanding of consumer behaviors in e-commerce live-streaming, research on consumer information search is limited. This study centers on the information search stage and sheds light on the determinants and moderators influencing consumers' information searching.

### 2.2. Consumer Online Information Search and Its Antecedents

Consumer information search refers to a process whereby consumers make inquiries within their social environment and obtain data to help them make a rational decision [25], including internal search and external search. Internal search refers to consumers' recalling relevant information from their memory based on past purchasing experience, while external search refers to consumers' active search for relevant information from the external environment, such as advertising, friends, or the observations of others [26]. Consumers can conduct an external search through online and offline channels [27]. Online information search is performed by asking family, friends, and consumer service through WeChat, QQ, etc., browsing shopping websites (i.e., Taobao, JD, Vipshop, etc.), and reading online consumer reviews on shopping platforms or other review-sharing websites (i.e., Xiaohongshu, TikTok, etc.). Consumers do not rely on a single channel when searching for information, but rather combine multiple information sources [28,29]. This study focuses on consumers' online information search, an external search, which refers to consumers' search for relevant product information in response to purchase demands from online channels during e-commerce live-streaming.

Scholars in consumer behavior research have examined the antecedents of information search behavior, including product characteristics, environmental factors, and individual traits [13,30,31]. Product involvement, as a product characteristic, is a critical motivator for information searching. For example, Santos and Goncalves [13] argue that high levels of dimensions regarding cognitive and affective involvement determine search with mobile

devices. Rokonuzzaman et al. [30] indicates that product involvement affects store loyalty through information search. Several scholars focus on the impact of environmental factors, such as time pressure and social media richness. For instance, in the offline context, Maity et al. [32] show that perceived risk, time pressure, involvement, and prior experience affect consumers' information search. In the online shopping environment, it has been confirmed that social media richness plays an important role in consumer information searching [33,34]. Some individual factors, such as consumers' education background and internet experience, also have a significant effect on their information searching [33,35]. Although the above studies strengthen our understanding of consumer information searching, they generally treat product characteristics, environmental factors, and individual factors separately, without delving into their interaction effects.

In this study, the COM-B model of behavior was utilized as a framework to elucidate the information search behavior of consumers in the context of e-commerce live-streaming. This model outlines how three key factors, namely capability, opportunity, and motivation, shape individual behavior [36]. Capability refers to the psychological and physical capacity of an individual to engage in a particular behavior. In the context of live-streaming, it pertains to the ability of consumers to effectively navigate and utilize online resources to find the desired information. However, in the current era of big data, consumers generally possess the capability to engage in information searching without significant constraints. Therefore, the research model did not explicitly incorporate factors related to consumers' capacity as it was assumed to be less of a limiting factor. Motivation encompasses the brain processes that drive and direct behavior. Motivation can be influenced by various factors such as pleasure, pain, hope, fears, social rejection, or social acceptance. Opportunity factors are the internal or external circumstances that facilitate or prompt a particular behavior. In this study, we considered the environmental factor of limited-quantity scarcity and the individual factor of impulsiveness as potential opportunities, while product characteristic of involvement served as a fundamental motivation for consumers' information searching.

### 2.3. Product Involvement

Product involvement refers to an individual's interest in an object because of its inherent values, necessity, and interest [9]. It can be categorized into two types, namely cognitive involvement and affective involvement [10]. Cognitive involvement emphasizes reasoning and factual information, which are induced by utilitarian or cognitive motives. Affective involvement focuses on emotion and mood, which derive from value-expressive or affective motives.

Product involvement has received extensive attention in advertising and consumer behavior research [10,14,37–39]. For example, Drossos et al. [10] examine the impacts of product involvement on purchase intentions in mobile text advertising. Chavadi et al. [40] confirm that the interaction effects of endorsement type and product involvement significantly affect consumers' attitude towards the brand and purchase intentions. Smith et al. [37] suggest that cognitive involvement has a direct effect on the perceived usefulness and ease of use of online shopping. However, how product involvement (i.e., cognitive and affective involvement) influences information processing and, especially, whether contextual variables, such as promotion strategy and individual characteristics, moderate the impact of product involvement on online information search still needs further investigation.

### 2.4. Online Scarcity Promotion Strategy

Scarcity means that "something is useful but of limited quantity" [41]. There are two types of scarcity, namely product scarcity and resource scarcity. Product scarcity refers to a lack of quantity, and resource scarcity refers to a lack of capital (whether financial, social, or cultural) and production inputs (i.e., time) [42]. Limited-quantity and limited-time scarcity are usually used as promotion strategies in practice [18,43]. Limited-quantity scarcity means that the available number of promotional products is limited, while limited-

time scarcity means that the promotion products are available for a predefined time, after which they are unavailable. Compared with limited-time scarcity, consumers face higher uncertainty in limited-quantity scarcity conditions because the consumer must compete against other buyers in addition to meeting the deadline set by the seller. E-commerce live-streamers usually use limited-quantity scarcity as a promotion strategy.

Scarcity can build a sense of urgency and tension among users and result in more purchases and shorter searches. A few scholars have demonstrated the impact of scarcity on consumers' impulse purchasing [44,45] and suggest that time scarcity may reduce consumers' possibility to make an accurate and objective judgment [46]. Aggarwal et al. [47] suggest that limited-quantity scarcity is an effective way to affect consumers' offline purchase intentions. Among these studies, most of them were confined to analysis of the immediate relationship between time scarcity and consumer behavior. We extend these studies in another direction by contending that limited-quantity scarcity may act as a moderator that weakens the product involvement–online information search link.

### 3. Hypothesis Development

#### 3.1. Product Involvement and Consumer Online Information Search

High-involvement products are more important and may induce more consequences if a poor decision is made, and thus consumers are more likely to devote their efforts to engage in cognitive activities and information searching to minimize risks [12]. Consumers make careful purchasing decisions when the products are important to their needs and values [48]. In general, high product involvement promotes consumers' engagement in information seeking, processing, as well as complicated decision behaviors.

Product involvement can be categorized into two types, namely cognitive involvement and affective involvement. High cognitive involvement means that consumers pay more attention to the functional and utilitarian aspects of products. For a product with high cognitive involvement, in order to gain more information about the function and utilitarian of the product, consumers are more likely to use third-party reviews, search engines, and competitors' product pages to search information [49]. Similarly, high affective involvement means that consumers pay more attention to the emotional aspects of products. Then, consumers are more likely to employ social media and this product's pages to search information and make a decision [49]. Thus, we propose the following hypotheses.

**H1a.** *Compared with low cognitive involvement, high cognitive involvement leads to higher consumer online information search.*

**H1b.** *Compared with low affective involvement, high affective involvement leads to higher consumer online information search.*

#### 3.2. The Role of Limited-Quantity Scarcity in Product Involvement–Information Searching

In the e-commerce live-streaming context, the limited-quantity scarcity offer is restricted to a set number of units. Every time a consumer buys a unit, the remaining number of units available will fall. This creates a sense of urgency and time pressure among purchasers. In such a situation, consumers lack time to consider the product's functions and values through information searching, and they are likely to apply some heuristic rule [27] using an information-filtering strategy for decision making, for instance, focusing on key product information. Under the same level of product involvement, consumers facing a high level of limited-quantity scarcity may be more likely to shorten searching time; that is, limited-quantity scarcity may weaken the relationship between product involvement and online information search. Thus, this study proposes the following hypotheses.

**H2a.** *Limited-quantity scarcity moderates the relationship between cognitive involvement and information search; as limited-quantity scarcity increases, the positive association between cognitive involvement and online information search is weakened.*

**H2b.** *Limited-quantity scarcity moderates the relationship between affective involvement and information search; as limited-quantity scarcity increases, the positive association between affective involvement and online information search is weakened.*

### 3.3. The Role of Personal Impulsiveness in Limited-Quantity Scarcity-Product Involvement–Information Searching

Impulsiveness is an enduring disposition to act impulsively in a consumer context, and it is a personal trait [50]. Individuals with this characteristic are more sensitive to environmental cues and often make decisions without thinking. In e-commerce livestreaming, consumers with high impulsiveness are likely to be more sensitive to limitedquantity scarcity promotion. They may experience greater anxiety and worry when they purchase a product in a limited-quantity promotion, which may strengthen the mitigating effect of limited-quantity scarcity on the production involvement–information searching link. This indicates that the moderating effect of limited-quantity scarcity on the relationship between product involvement and information searching will be stronger for consumers with high impulsiveness. Thus, we formulate the following hypothesis.

**H3a.** *The negative moderating effect of limited-quantity scarcity on the relationship between cognitive involvement and information searching is stronger when consumers' impulsiveness is higher.*

**H3b.** *The negative moderating effect of limited-quantity scarcity on the relationship between affective involvement and information searching is stronger when consumers' impulsiveness is higher.*

The theoretical model of this research is depicted in Figure 1. Drawing from COM-B behavior changing theory, product involvement was considered as motivating factor, and limited-quantity scarcity and impulsiveness were selected as opportunity factors. By investigating the moderating effect of limited-quantity scarcity and impulsiveness, the model aims to show the boundary conditions regarding the effect of product involvement on consumer information search.

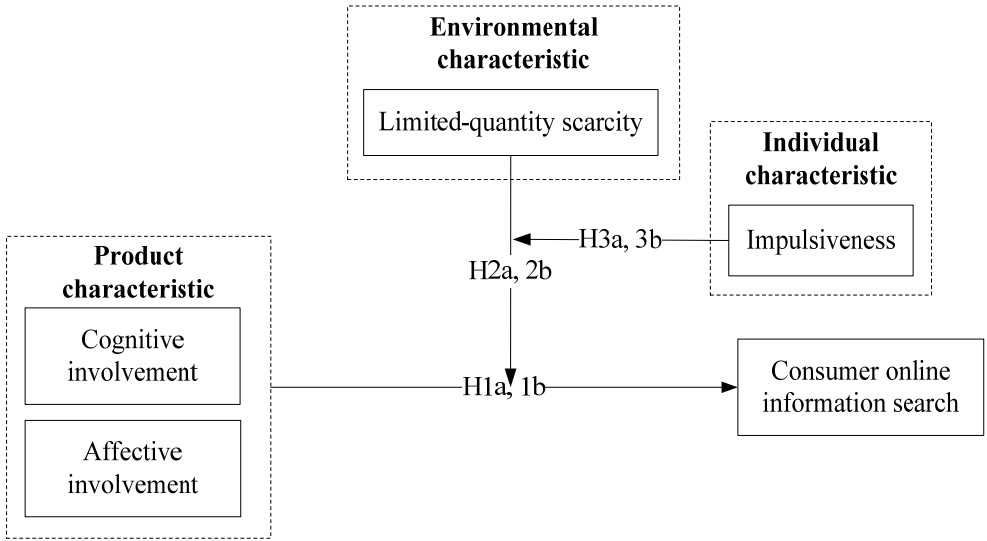

**Figure 1.** Theoretical model.

## 4. Research Methodology
### 4.1. Experimental Design, Participants, and Procedure

A survey experiment was conducted to empirically validate the model. It was conducted by sending a different version of the questionnaire to each group of respondents [51]. The experiment in this study used a 2 (cognitive involvement: high vs. low) × 2 (affective

involvement: high vs. low) × 2 (limited-quantity scarcity: high vs. low) between-subjects design. Thus, eight versions of the questionnaire were designed. The respondents were then randomly assigned to one of the above conditions.

The independent variables (i.e., cognitive involvement and affective involvement) were manipulated through selected product categories. First, based on Drossos et al. [10] and considering Chinese consumption habits, we selected four products that represent different degrees of cognitive and affective involvement: a vacuum-insulated cup, chocolate, an electric mosquito repellent, and mineral water. The four products satisfy the following criteria: (1) each represents a combination of different levels of cognitive and affective involvement; (2) each is a daily necessity closely related to the participants; (3) the price of each product is under CNY 100; and (4) all are frequently purchased non-durable consumer items.

The e-commerce live-streams for each product were designed in the form of text and pictures. To control the impact of product brand on consumer online information search, fictional brands were used in this study. The questionnaire started with a screening question to make sure that the respondents had experience watching e-commerce live-streaming. Then, the following scenario was presented to the respondents: "Imagine that you are watching an e-commerce live-stream (for example, Li Jiaqi, Xin Youzhi, Luo Yonghao, or other live-streamers). The live-streamer is explaining the JINGGUANG vacuum cup, and the description of the product is as follows:

There are four styles of this vacuum-insulated cup, and six colors are available. The smaller vacuum-insulated cup holds 400 mL, and the larger one holds 450 mL. The bottle's inner wall is made of 304/316 stainless steel, which keeps your favorite beverage hot or cold for hours. BPA free, nontoxic, and eco-friendly. In offline stores, the large one is 89 RMB, and the small one is 79 RMB. Today's live room benefits: please grab a red envelope before placing an order, and you will get the big cup for 59 RMB and the small cup for 49 RMB! Only 100 items left in stock! Once they are gone, this product will no longer be sold! (Or Sufficient stock! Please buy it without worry!).

After the scenario, the respondents need to indicate their online information search plans and to complete the manipulation check measures. Manipulation checks are carried out to verify whether the experimental stimuli were successful. Finally, demographic variables were collected, such as gender, age, and purchase frequency.

*4.2. Pre-Test: Product Selection*

A preliminary test with 105 participants was conducted to determine whether the four products could represent the four experimental conditions. We made some modifications based on the scale of Putrevu and Lord [52] to measure the cognitive and affective involvement of each product (see Table 1). Specifically, we utilized a seven-point Likert scale to assess the extent to which consumers' decision to purchase a product is influenced by cognitive or affective involvement. Subsequently, we computed the average scores for cognitive involvement (CI) and affective involvement (AI) for each product. These mean scores were then used to evaluate the effectiveness of the cognitive and affective involvement manipulations within the chosen product categories. The pre-test results showed that all of the products could represent the corresponding cognitive and affective involvement, except for chocolate [Mean $_{chocolate}$ (CI) = 5.61, Mean $_{chocolate}$ (AI) = 5.24], which is expected to represent high-level affective involvement and low-level cognitive involvement. However, the results indicate that consumers' decision to purchase chocolate is affected by both high-level cognitive and high-level affective involvement. This may be due to the different eating habits of foreigners and Chinese. Thus, we replaced the chocolate with duck neck; the results showed that duck neck [Mean $_{duck neck}$ (CI) = 4.055, Mean $_{duck neck}$ (AI) = 5.776] was more representative than chocolate. Finally, we selected a vacuum-insulated cup, duck neck, electric mosquito repellent, and mineral water as the experimental products (please refer to Table 1).

**Table 1.** Four experimental products.

| | | Cognitive Involvement (CI) | |
|---|---|---|---|
| | | **High** | **Low** |
| Affective involvement (AI) | High | Vacuum-insulated cup | Duck neck |
| | Low | Electric mosquito repellent | Mineral water |

### 4.3. Measures

To ensure content validity, the construct items for all variables and manipulation items were derived from validated scales in the extant literature. In the meantime, they were reworded to be more suitable for our research context. The manipulation measures for cognitive involvement and affective involvement were adapted from Putrevy and Lord [52]. Manipulation for limited-quantity scarcity was measured by asking respondents a question using a seven-point Likert scale (i.e., Do you think that the quantity of vacuum-insulated cup inventory provided by the live-streamer is sufficient?). The measures for consumer online information search were adapted from Murray [26]. For impulsiveness, we adapted the measures from Rook and Fisher [50]. We used a seven-point Likert scale to measure all the items. Four academic and industry experts were invited to review the questionnaire. Based on their suggestions, we improved several measurement items to ensure that they were easy to understand. Table A1 shows the final measurement items (see Appendix A).

### 4.4. Data Collection

We distributed the eight versions of the questionnaire to 483 respondents via wjx.cn. We eliminated 81 invalid questionnaires, as follows: (1) surveys by participants who did not have experience of e-commerce live-streaming, (2) surveys with blank answers, (3) surveys in which all items had the same answer, and (4) cases in which the respondents took too little time (e.g., less than 90 s) to fill in the questionnaire. We totally collected 402 valid questionnaires.

## 5. Data Analysis and Results

### 5.1. Reliability and Validity Analysis

We used an exploratory factor analysis (EFA) to test the validity of the questionnaire [53]. We first conducted Bartlett's test of sphericity and used the Kaiser–Meyer–Olkin (KMO) measure of sampling adequacy to test the appropriateness of the sample for the EFA. As depicted in Table 2, the KMO values were greater than the threshold of 0.60 ($KMO_{cognitive\ involvement}$ = 0.795, $KMO_{affective\ involvement}$ = 0.843, $KMO_{consumer\ information\ search}$ = 0.703, $KMO_{impulsiveness}$ = 0.792), and the results for Bartlett's sphericity test were significant ($p < 0.05$), indicating that we can further conduct the EFA. Second, using the principal component analysis with varimax rotation, we conducted exploratory factor analysis [53]. As shown in Table 3, the results showed that the minimum item loading was 0.808, and all of them were more than the threshold of 0.5. The minimum cumulative interpreted variance was 71.366%, which is over the threshold level of 60%. Based on the above results, we can conclude that all the measurements had a good validity. Moreover, the minimum Cronbach's $\alpha$ was 0.795, indicating that all the constructs have a good reliability.

**Table 2.** KMO and Bartlett's test.

| | Cronbach's $\alpha$ | KMO | Bartlett Sphericity Test | | | Cumulative Interpreted Variance |
|---|---|---|---|---|---|---|
| | | | **Approximate Chi-Square Value** | **df** | **Sig.** | |
| Consumer information search | 0.795 | 0.703 | 375.529 | 3 | 0.000 | 71.366% |
| Cognitive involvement | 0.903 | 0.795 | 1094.258 | 6 | 0.000 | 77.591% |
| Affective involvement | 0.955 | 0.843 | 2389.654 | 10 | 0.000 | 85.090% |
| Impulsiveness | 0.867 | 0.792 | 799.609 | 6 | 0.000 | 71.623% |

**Table 3.** Rotated component matrix.

| Variables | AI | CI | IMP | CIS |
|:---:|:---:|:---:|:---:|:---:|
| CIS1 | 0.146 | 0.108 | −0.102 | **0.808** |
| CIS2 | −0.043 | 0.255 | −0.03 | **0.808** |
| CIS3 | 0.105 | 0.265 | 0.052 | **0.822** |
| CI1 | 0.147 | **0.871** | 0.034 | 0.199 |
| CI2 | 0.064 | **0.845** | −0.054 | 0.235 |
| CI3 | 0.159 | **0.871** | −0.025 | 0.142 |
| CI4 | 0.131 | **0.823** | 0.008 | 0.135 |
| AI1 | **0.923** | 0.015 | −0.018 | −0.044 |
| AI2 | **0.907** | 0.164 | −0.153 | 0.091 |
| AI3 | **0.943** | 0.16 | −0.096 | 0.086 |
| AI4 | **0.884** | 0.188 | 0.032 | 0.136 |
| IMP1 | −0.053 | 0.021 | **0.802** | 0.072 |
| IMP2 | −0.058 | −0.075 | **0.884** | 0.018 |
| IMP3 | −0.019 | −0.062 | **0.86** | −0.154 |
| IMP4 | −0.064 | 0.076 | **0.828** | −0.045 |

Note: Bold numbers indicate outer loading on the assigned constructs; CIS = consumer information search; AI = affective involvement; CI = cognitive involvement; IMP = impulsiveness.

### 5.2. Manipulation Checks

An independent sample *t*-test was used to test whether the variables were successfully manipulated. As Table 4 shows, the four selected products represent the corresponding product involvement. The difference in cognitive involvement between the vacuum-insulated cup [Mean (CI) = 5.71] and duck neck [Mean (CI) = 4.055] is significant at $p < 0.05$, and the differences between the other different products are significant at $p < 0.01$. This finding suggests that the manipulations of the involvement constructs through the selected product categories were successful. As depicted in Table 5, there are also significant differences in the mean values of limited-quantity scarcity between the high limited-quantity group and low limited-quantity group (t = 13.87, $p < 0.01$), indicating that limited-quantity scarcity was successfully manipulated.

### 5.3. Hypothesis Testing

5.3.1. Main Effects of Cognitive Involvement and Affective Involvement

We employed an independent sample t-test to test the effects of the two types of product involvement on online information search. The results show (Table 6) that cognitive involvement had a significant influence on online information search (t = 4.414, $p = 0.000 < 0.01$), indicating that the respondents were more likely to conduct online information search for high-cognitive-involvement products than low-cognitive-involvement products. Thus, H1a was supported. However, the t-test analysis showed that the impact of affective involvement on online information search was not significant (t = 0.02, $p = 0.984 > 0.1$). Thus, H1b was not supported.

5.3.2. The Effect of Limited-Quantity Scarcity on Product Involvement–Information Searching

The multivariate analysis of variance (MANOVA) was conducted to examine the moderating effects of limited-quantity scarcity and impulsiveness. As shown in Table 7, the moderating effect of limited-quantity scarcity between cognitive involvement and online information search was significant (F = 12.978, $p = 0.000 < 0.05$). Thus, H2a was supported. However, the interaction effect of limited-quantity scarcity, and the affective involvement

on online information search was not significant (F = 1.449, $p$ = 0.229 > 0.05). Therefore, H2b was not supported.

**Table 4.** The manipulation check for cognitive involvement and affective involvement.

| Products | Mean | |
|---|---|---|
| | **Cognitive Involvement (CI)** | **Affective Involvement (AI)** |
| Vacuum-insulated cup (code = 1) | 5.71 (high) | 5.97 (high) |
| Duck neck (code = 2) | 4.06 (low) | 5.78 (high) |
| Electric mosquito repellent (code = 3) | 5.60 (high) | 3.94 (low) |
| Mineral water (code = 4) | 3.98 (low) | 3.91 (low) |
| Independent samples test for cognitive involvement (sig.) | | Mean difference |
| Difference between 1 and 2 | $p < 0.05$ (t = 6.85) | 1.66 |
| Difference between 3 and 2 | $p < 0.01$ (t = 7.35) | 1.54 |
| Difference between 1 and 4 | $p < 0.01$ (t = 6.21) | 1.74 |
| Difference between 3 and 4 | $p < 0.01$ (t = 6.42) | 1.62 |
| Independent samples test for affective involvement (sig.) | | Mean difference |
| Difference between 1 and 3 | $p < 0.01$ (t = 8.34) | 2.04 |
| Difference between 1 and 4 | $p < 0.01$ (t = 8.19) | 2.06 |
| Difference between 2 and 3 | $p < 0.01$ (t = 7.61) | 1.84 |
| Difference between 2 and 4 | $p < 0.01$ (t = 7.48) | 1.86 |

**Table 5.** The manipulation check for limited-quantity scarcity.

| Products | Mean | | Sig. | T-Value |
|---|---|---|---|---|
| | **Limited-Quantity Scarcity** | | | |
| Vacuum-insulated cup (code = 1) | 3.48 (high) | 5.72 (low) | $p < 0.01$ | 6.16 |
| Duck neck (code = 2) | 3.04 (high) | 5.56 (low) | $p < 0.01$ | 7.22 |
| Electric mosquito repellent (code = 3) | 3.32 (high) | 5.76 (low) | $p < 0.01$ | 7.72 |
| Mineral water (code = 4) | 3.20 (high) | 5.56 (low) | $p < 0.01$ | 6.54 |
| Total | 3.26 (high) | 5.65 (low) | $p < 0.01$ | 13.87 |

**Table 6.** The main effect of cognitive involvement and affective involvement on consumer online information search.

| Dependent Variable | Independent Variable | Mean | Std | T-Value | Sig. |
|---|---|---|---|---|---|
| | CI (high) | 5.49 | 1.08 | 4.41 | 0.000 ** |
| Consumer online information search | CI (low) | 4.80 | 1.13 | | |
| | AI (high) | 5.15 | 1.00 | 0.02 | 0.984 |
| | AI (low) | 5.15 | 1.30 | | |

Notes: CI = cognitive involvement; AI = affective involvement; ** $p < 0.05$.

To probe more insight into the interaction effect, we followed the procedure of Aiken et al. [54] to decompose the interaction terms and plot the relationships in Figure 2. From Figure 2, it can be seen that compared to low-limited-quantity scarcity conditions, cognitive involvement had less impact on online information search under high-limited-quantity scarcity conditions. This result indicates that limited-quantity scarcity negatively moderates the relationship between cognitive involvement and online information search.

**Table 7.** The MANOVA analysis results on the moderating effect of limited-quantity scarcity.

| Source | Mean Square | F-Value | Sig. | $R^2$ |
|---|---|---|---|---|
| CI | 47.610 | 46.804 | 0.000 ** | |
| LQS | 67.788 | 66.640 | 0.000 ** | 0.242 |
| CI × LQS | 13.201 | 12.978 | 0.000 ** | |
| Error | 1.017 | | | |
| AI | 0.001 | 0.001 | 0.975 | |
| LQS | 67.788 | 58.111 | 0.000 ** | 0.131 |
| AI × LQS | 1.690 | 1.449 | 0.229 | |
| Error | 1.167 | | | |

Notes: CI = cognitive involvement; AI = affective involvement; LQS = limited-quantity scarcity; ** $p < 0.05$.

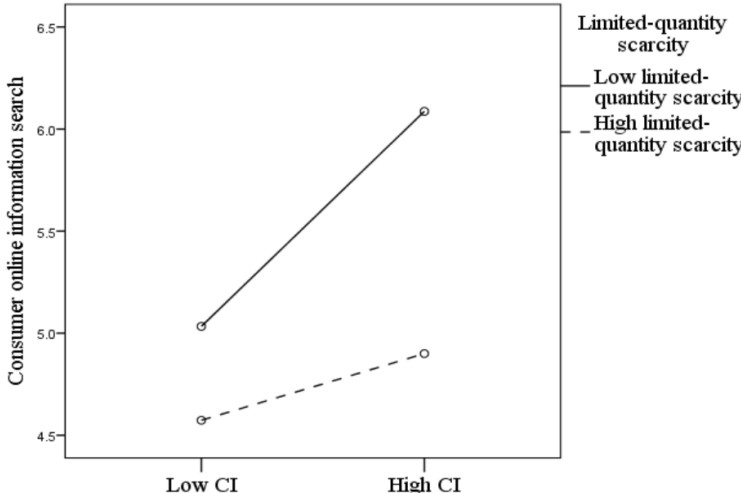

Notes: CI = cognitive involvement.

**Figure 2.** The interaction effect of cognitive involvement and limited-quantity scarcity.

### 5.3.3. The Three-Way Interaction Effects among Product Involvement, Limited-Quantity Scarcity, and Impulsiveness

The MANOVA analysis results on the moderating effect of personal impulsiveness are shown in Table 8. The results show that the three-way interaction among cognitive involvement, limited-quantity scarcity, and impulsiveness was statistically significant (F = 3.994, $p = 0.008 < 0.05$). At the same time, the three-way interaction among affective involvement, limited-quantity scarcity, and impulsiveness was also statistically significant (F = 4.164, $p = 0.006 < 0.05$). To further gain more insights into the moderating effects, we plotted the three-way interaction using simple slopes analysis following the procedure of Aiken et al. [54] and Dawson and Richter [55], and the relationships are plotted in Figures 3 and 4.

As shown in Figure 3a, for consumers with low impulsiveness, cognitive involvement was positively related to online information search whether limited-quantity scarcity is high or low. For another, as depicted in Figure 3b, for consumers with high impulsiveness, cognitive involvement was positively associated with information searching in situations of low limited-quantity scarcity. However, this positive relationship flattens in situations of high limited-quantity scarcity. This result indicates that the mitigating effect of high limited-quantity scarcity on the relationship between cognitive involvement and information searching is enhanced when individuals' impulsiveness is high. Therefore, H3a was supported.

**Table 8.** The MANOVA analysis results on the three-way interaction effects.

| Source | Mean Square | F-Value | Sig. | $R^2$ |
|---|---|---|---|---|
| CI | 44.878 | 45.354 | 0.000 ** | |
| LQS | 61.392 | 62.044 | 0.000 ** | |
| IMP | 3.118 | 3.151 | 0.077 | 0.270 |
| CI × LQS | 9.742 | 9.845 | 0.002 ** | |
| CI × LQS × IMP | 3.952 | 3.994 | 0.008 ** | |
| Error | 0.990 | | | |
| AI | 0.088 | 0.078 | 0.780 | |
| LQS | 62.309 | 54.889 | 0.000 ** | |
| IMP | 2.718 | 2.394 | 0.123 | 0.163 |
| AI × LQS | 2.432 | 2.143 | 0.144 | |
| AI × LQS × IMP | 4.727 | 4.164 | 0.006 ** | |
| Error | 1.135 | | | |

Notes: CI = cognitive involvement; AI = affective involvement; LQS = limited-quantity scarcity; IMP = impulsiveness; ** $p < 0.05$.

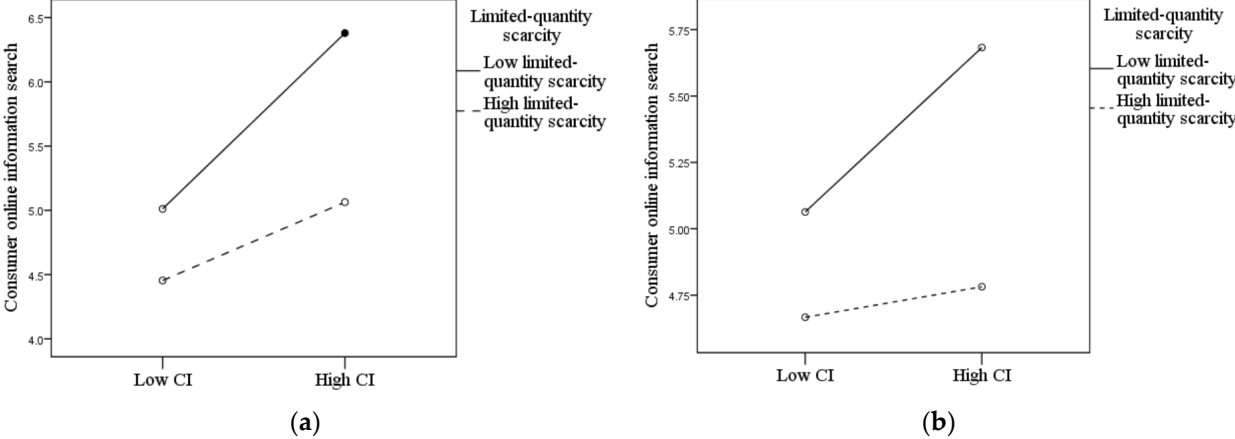

**Figure 3.** Effects of personal impulsiveness on LQS-CI-CIS. (**a**) Effects of low personal impulsiveness on LQS-CI-CIS. (**b**) Effects of high personal impulsiveness on LQS-CI-CIS. Notes: CI = cognitive involvement; LQS = limited-quantity scarcity; CIS = consumer information search.

As shown in Figure 4a, for consumers with low impulsiveness, affective involvement is positively related to information searching, and there is a smaller difference in the relationship between affective involvement and information searching under the conditions of high or low limited-quantity scarcity. Thus, limited-quantity scarcity does not moderate the relationship between affective involvement and information searching for consumers with low impulsiveness. Surprisingly, as shown in Figure 4b, for consumers with high impulsiveness, affective involvement is negatively related to information searching no matter whether limited-quantity scarcity is high or low. In the meantime, this negative impact is stronger in situations of high limited-quantity scarcity. This result indicates that high limited-quantity scarcity strengthens the negative impact of affective involvement on information searching for consumers with high impulsiveness. This result is contrary to the hypothesis. Thus, despite the significance of the three-way interaction effect, H3b was still not supported.

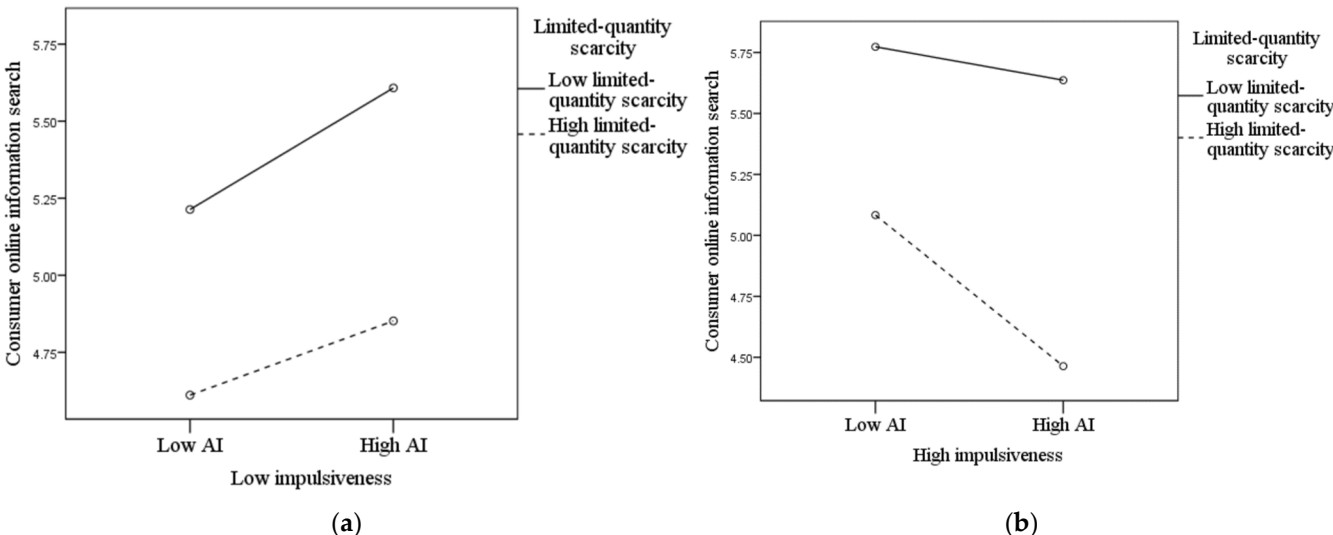

**Figure 4.** Effects of personal impulsiveness on LQS-AI-CIS. (**a**) Effects of low personal impulsiveness on LQS-AI-CIS. (**b**) Effects of high personal impulsiveness on LQS-AI- CIS. Notes: AI = affective involvement; LQS = limited-quantity scarcity; CIS = consumer information search.

## 6. Discussion

### 6.1. Discussion of the Results

Although a few studies have shown the impacts of product characteristics, environmental factors, and individual traits on consumer information search, these factors have been examined in isolation. The literature relating to the interaction effect of these factors on information searching is scant. Thus, this study examines the three-way interaction effects among limited-quantity scarcity, personal impulsiveness, and product involvement on consumer information search.

The results support four of six hypotheses. As expected, the results verify that cognitive involvement significantly affects online information search, and limited-quantity scarcity significantly weakened the relationship between cognitive involvement and online information search. This is consistent with the findings of the prior literature [18,56,57], which suggest that a scarcity-based promotion provides a sense of competition and stimulates consumers' excitement, which causes them to pay less attention to the cognitive functions of the products. More importantly, this study gains new findings by investigating how personal impulsiveness strengthens the mitigating effect of limited-quantity scarcity on the relationship between cognitive involvement and information searching.

It is worth noting that affective involvement did not significantly influence consumer online information search, and the interaction effect of affective involvement and limited-quantity scarcity on information searching was also not significant. This is consistent with the findings of Schaefer et al. [11], suggesting that cognitive involvement influences search activities by activating the left BA44 of brain, but affective involvement has no effect on search activities.

Interestingly, it has been proved that limited-quantity scarcity had strikingly different regulating effects for consumers with different impulsiveness. Specifically, for consumers with low impulsiveness, affective involvement is positively related to information searching no matter whether limited-quantity scarcity is high or low. In the meantime, there was a smaller difference in the influence of affective involvement on information searching in situations of high and low limited-quantity scarcity. This means that limited-quantity scarcity does not moderate the relationship between affective involvement and information searching for consumers with low impulsiveness. However, for consumers with high impulsiveness, affective involvement is negatively related to information searching no matter whether limited-quantity scarcity is high or low, and this negative impact is stronger

in situations of high limited-quantity scarcity. This means that limited-quantity scarcity strengthens the negative impact of affective involvement on information searching for consumers with high impulsiveness. The reason for this may be that affective involvement is induced by emotions and feelings, and affective cues play the most prominent role in the formation of purchase decisions. For consumers with high impulsiveness, they are more sensitive to environmental cues and often make decisions without thinking. Thus, when they like a product with high affective involvement, they are more likely to make purchase decisions based on their affect and feelings and do not want to think extensively or search for more information, especially under conditions with high limited-quantity scarcity.

### 6.2. Theoretical Implications

This study has some important theoretical implications. First, rather than repeat previous investigations of impulse purchases and purchase intentions, this paper focuses on pre-purchase online information searching in the context of e-commerce live-streaming. Although previous studies reached the consensus that product involvement is related with information search [30,32,58], the independent effects of cognitive and affective involvement on online information search are limited. This study distinguishes the impacts of cognitive and affective involvement on information search and finds that high cognitive involvement will lead to high online information search, whereas affective involvement is not related with online information search. The findings of this study may deepen existing knowledge about consumer information search and encourage future researchers to pay attention to the differences between cognitive and affective involvement in e-commerce live-streaming research.

By highlighting the impact of limited-quantity scarcity as a moderating variable, this study expands our understanding of limited-quantity scarcity strategy in e-commerce live-streaming. While prior research confirmed the direct influence of limited-scarcity promotion on consumers' purchase decision making [18], this study extends existing research by uncovering the mechanisms and process of limited-scarcity strategy in affecting consumer information search. This provides a more comprehensive understanding of how consumers make purchasing decisions in the context of scarcity promotion strategy.

More importantly, since consumers' information search behavior is the result of multiple factors, including product characteristics, environmental factors, and individual traits, it is necessary to take account of the simultaneous interaction effects among these factors. By exploring the three-way interaction among product involvement, limited-quantity scarcity, and personal impulsiveness on online information search, the findings of this study enrich the existing literature and enable us to accurately predict consumers' searching behavior in real business. This study also provides an explanation for the findings of previous studies arguing that cognitive involvement is significantly related to search behavior [11], while affective involvement is not associated with search level by highlighting the combined moderating effect of limited-quantity and personal impulsiveness.

### 6.3. Practical Implications

The findings of this study could help reduce unsustainable information search behavior and promote the sustainability of live e-commerce. This study focuses on the pre-purchase stage of consumer information search behavior in e-commerce live-streaming. Although retailers have used a variety of promotion strategies to stimulate consumers to make quick decisions and increase impulse purchases [18,47,59], information searching is still an essential stage in the consumer decision-making process. Understanding the determinants of information searching is essential for businesses to manage their external information source and internal website information, thereby providing usable information to consumers for decision making and promoting the sustainable and healthy development of live-streaming. This study has verified the positive relationship between cognitive involvement and online information search. Therefore, retailers and live-streamers selling high-cognitive-involvement products should pay close attention to consumers' information

searches and try to communicate quality information and ensure product quality to reduce consumers' perceived risk, promoting a sustainable and thriving live economy. Retailers should also remember that it takes time and energy to search for information, and thus they should make information available and accessible to consumers, thereby promoting consumers' effective information searching and developing a sustainable-consumption mindset and reducing the likelihood of unsustainable information searching. For example, in addition to providing necessary product information during e-commerce live-streaming, other platforms related to product marketing should also display relevant product information and consumer reviews. Retailers should try to maintain users' online reviews across multiple platforms.

Second, limited-quantity scarcity has been used to attract potential consumers and promote impulse purchases in e-commerce live-streaming. The findings of this study remind retailers and streamers to pay attention to the differences between cognitive and affective product by arguing that the limited-quantity scarcity strategy significantly weakens the impact of cognitive involvement on online information search, while it does not have a moderating effect on the affective involvement–online information search link. Thus, retailers should realize that scarcity promotion strategies are more effective in mitigating the effect of product involvement for products with high cognitive involvement than products with high affective involvement, thus helping retailers apply limited-quantity scarcity strategy effectively and contributing to the sustainable development of the live-streaming economy.

Third, retailers should recognize that the negative effect of limited-quantity scarcity on the positive relationship between product involvement on online information search is contingent upon consumers' impulsiveness. This means that online retailers need to design customized scarcity promotions that target different segments of online consumers across personal impulsiveness. Thus, consumers should reasonably examine their impulsiveness and establish the concept of sustainable consumption to reduce the possibility of irrational buying decisions.

## 7. Conclusions, Limitations, and Future Research

This study develops a research model to examine the moderating effect of limited-quantity scarcity and impulsiveness on the relationship between product involvement and information searching. The results indicate that (1) cognitive involvement is positively associated with online information search, whereas affective involvement is not associated with online information search; (2) limited-quantity scarcity significantly weakens the impact of cognitive involvement on online information search, but it does not have an interaction effect with affective involvement on online information search; and (3) the three-way interaction among product involvement (i.e., cognitive involvement, affective involvement), limited-quantity scarcity, and impulsiveness on consumer information search is significant.

A few limitations and future directions should be discussed. First, most of the respondents were under the age of 30. Although it has been reported that 78% of people who watch live-streaming are under the age of 30 [60], future studies could try to include more samples with a wider range of ages to strengthen the robustness of our conclusions. In addition, all the samples were collected from China, which may limit the generalizability of the findings. Shobeiri et al. [61] suggested that the experiential value of e-retailing websites influences North American versus Chinese consumers differently. In light of this, consumers' perceptions and reactions towards e-commerce live-streaming, such as scarcity promotion strategies, can vary across culture. Therefore, future studies should explore cultural differences in perceptions of and reactions to e-commerce live-streaming. To make the findings more reliable and representative, it is recommended that future research extend the surveys to include consumers from other countries, particularly those in Western contexts. Second, a picture and text were used to induce participants to imagine the e-commerce live-streaming situation, which may weaken the influences of product involve-

ment and limited-quantity scarcity. Therefore, future studies can use real live-streaming in the form of video to validate our results. Third, the use of a single product to represent each quadrant of the cognitive and affective involvement grid limits the generalization of this study, especially for those products that cannot be easily classified into any or only one quadrant. Thus, different products can be considered to verify this conclusion in the future. Furthermore, it would be interesting to explore whether the findings of this study can be extended to intangible service contexts. The focus of attention could shift towards examining how involvement influences information search behavior when it comes to acquiring services rather than physical goods, aiming to identify potential similarities and differences.

**Author Contributions:** Conceptualization, Y.G. and C.W.; methodology, X.C.; validation, X.C. and Y.G.; formal analysis, X.C.; investigation, X.C.; resources, C.W.; data curation, X.C.; writing—original draft preparation, Y.G. and C.W.; writing—review and editing, Y.G. and C.W.; visualization, X.C.; supervision, C.W.; project administration, Y.G. and C.W.; funding acquisition, C.W. All authors have read and agreed to the published version of the manuscript.

**Funding:** This research was funded by National Natural Science Foundation of China [71701034] and Fundamental Research Funds for the Central Universities of China (Dalian Maritime University) [3132022277].

**Informed Consent Statement:** Informed consent was obtained from all subjects involved in the study. Written informed consent from the participants was not required to participate in this study in accordance with the national legislation and the institutional requirements.

**Data Availability Statement:** The data presented in this study are available on request from the corresponding author. The data are not publicly available due to privacy restrictions.

**Conflicts of Interest:** The authors declare no conflict of interest.

## Appendix A. Measurement Items

**Table A1.** Questionnaire Items.

| Construct | Items | Reference |
|---|---|---|
| Consumers' online information search | 1. I will search for relevant information about the product again on other shopping platforms (such as Taobao, JD, Pinduoduo, etc.) before making purchase decisions.<br>2. I will consider what a search engine (such as Baidu.com, Google.com, etc.) says about the product.<br>3. I will refer to some of the reviews of the website (such as: public comment, Xiao Hong Book, etc.) to make my purchase decision. | [26] |
| Cognitive involvement | 1. Whether to buy is mainly based on functional facts.<br>2. I need to consider a lot of factors when I make a purchase decision.<br>3. The function of the product is very important to me.<br>4. If not carefully thought out, the features of this product may be very poor. | [52] |
| Affective involvement | 1. Whether to buy is mainly based on looks/taste/touch/smell/sounds.<br>2. My purchase decision is based on whether I like this product.<br>4. My purchase decision is based on my feelings on this product. | [52] |
| Impulsiveness | 1. I often buy things spontaneously.<br>2. I often buy things without thinking.<br>3. I will buy it quickly when I see something I like. | [50] |

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
