# Peer review of "Consumer Information Search in Live-Streaming: Product Involvement and the Moderating Role of Scarcity Promotion and Impulsiveness"

_sustainability, doi:10.3390/su151411361_

Round 1

Reviewer 1 Report

Dear Authors,

I read your article with great interest. Can you explain how to generalize the results in the Chinese context and worldwide? As well, please explicitly describe the validity of the theoretical model. 

The language is easy to read and understand. Minor editing is needed.

Author Response

Thank you very much for your support and for your careful reading of the manuscript. We really appreciate your insightful comments that have significantly improved the manuscript.

Reviewer 2 Report

The article investigates the process of information acquisition in live streaming shopping. The article is very well organized, clear and is informative. The theoretical background is adequate, comprehensive and clear. The list of references is also adequate. The methods employed is suitable and the discussion is comprehensive too. No major shortfalls have been identified. I only have three points for consideration: First, please simplify the title of the paper (i.e. shorten the title), and second, be clear about the meaning of product involvement from the first beginning (from the abstract section). The topic is appealing and is important and if you clarify the meaning of product involvement at the beginning, it will be more encouraging to complete reading the paper to the end. Finally, it would be value adding if the authors discuss the same concept from the service perspective. In other words, does the same concepts and ideas presented apply for intangible services or are they exclusive to physical (tangible) product marketing/purchasing? What are the similarities and differences in this regard. Do the same findings apply equally on both items; products and services? Another minor issue to consider. How likely different customers from different contexts be inclusive in the research findings. Do you think these findings apply at an international level? Please address this issue. The more you extend the scope of this study, the more it will be value adding.

There are several typos that have been spotted. Kindly revise the whole paper in order to fix. 

Author Response

Dear Reviewer:

Thank you very much for your support and for your careful reading of the manuscript. We really appreciate your insightful comments that have significantly improved the manuscript.

Sincerely,

The Authors

Reviewer 3 Report

In a digital society, the analysis of consumer behavior in e-commerce is welcome.

The existence of a preliminary test for product identification could represent the four experimental conditions​​​​​​​, as well as the rigorous analysis of the results of the questionnaire applied by the authors shows their attention to the processing of the obtained data.

However, the results and conclusions obtained by the authors must also emphasize the fact that they worked with variables on the ordinal scale. Coding the responses of the variables using the Likert scale does not mean that, based on the results obtained, all statistical indicators related to the numerical variables (for example "mean") can be calculated. Therefore, it is preferable to use the "median" and the "mode" as indicators of the central tendency.

If possible, I would like an explanation of the "mean" indicator in this situation.

As a suggestion for further research, as the authors stated that ​​​​​​​Likert scale was used to quantify the responses, ordinal regression models or logistic regression or non-parametric analysis methods can be used. 

Author Response

(The authors gave the same response as above.)

Reviewer 4 Report

The overall content of the manuscript is easy to understand, however, there are some area of improvement, as follows:

1.     Any evidence for the past work on line 76?

2.     Some formatting issue, examples as follows: L39 – space between comma (,) and interactivity; L95 – space between on and consumers’; L98 – space between the dot (.) and Lu. Please check the whole manuscript. 

3.     Section 2.1 – Please state, as conclusion of the section, which theory is the basis for the research.

4.     Section 2.2 – First paragraph – confusion with the term consumer and customers. L121 – consumer information search definition use the term customers (L121 and L 134), which contradicted with the term consumers (L131). Please also check the whole manuscript, either to apply consumer or customer.

5. Figure 1 (Theoretical Model) L252 - Perhaps to properly explain how the model emerged.

Author Response

(The authors gave the same response as above.)

Round 2

Reviewer 2 Report

All the required changes have been made to the satisfaction of the reviewer. Good luck in your publication.